

# Surface and subsurface Labrador Shelf water mass conditions during the last 6,000 years

Annalena A. Lochte[1,2*], Ralph Schneider[1], Janne Repschläger[3], Markus Kienast[4], Thomas Blanz[1], Dieter Garbe-Schönberg[1], Nils Andersen[5]

[1]Institute of Geoscience, University of Kiel, Ludewig-Meyn Str. 10, 24118 Kiel, Germany
[2]GEOMAR Helmholtz-Centre for Ocean Research Kiel, Wischofstraße 1-3, 24148 Kiel, Germany
[3]Max-Planck Institute for Chemistry, Hahn Meitner Weg 1, 55128 Mainz, Germany
[4]Department of Oceanography, Dalhousie University, 1355 Oxford Street, Halifax, Canada
[5]Leibniz Laboratory for Radiometric Dating and Stable Isotope Research, University of Kiel, Max-Eyth-Str. 11–13, 24118 Kiel, Germany

*Correspondence to*: Annalena A. Lochte (annalena.lochte@ifg.uni-kiel.de)

**Abstract.** The Labrador Sea is important for the modern global thermohaline circulation system through the formation of intermediate Labrador Sea Water (LSW) that has been hypothesized to stabilize the modern mode of North Atlantic deep-water circulation. The rate of LSW formation is controlled by the amount of winter heat loss to the atmosphere, the expanse of freshwater in the convection region and the inflow of saline waters from the Atlantic. The Labrador Sea, today, receives freshwater through the East and West Greenland Currents (EGC, WGC) and the Labrador Current (LC). Several studies have suggested the WGC to be the main supplier of freshwater to the Labrador Sea, but the role of the southward flowing LC in Labrador Sea convection is still debated. At the same time, many paleoceanographic reconstructions from the Labrador Shelf focussed on late Deglacial to early Holocene meltwater run-off from the Laurentide Ice Sheet (LIS), whereas little information exists about LC variability since the final melting of the LIS about 7,000 years ago. In order to enable better assessment of the role of the LC in deep-water formation and its importance for Holocene climate variability in Atlantic Canada, this study presents high-resolution middle to late Holocene records of sea surface and bottom water temperatures, freshening and sea ice cover on the Labrador Shelf during the last 6,000 years. Our records reveal that the LC underwent three major oceanographic phases from the Mid- to Late Holocene. From 6.2 to 5.6 ka BP, the LC experienced a cold episode that was followed by warmer conditions between 5.6 and 2.1 ka BP, possibly associated with the late Holocene Thermal Maximum. Although surface waters on the Labrador Shelf cooled gradually after 3 ka BP in response to the Neoglaciation, Labrador Shelf subsurface/bottom waters show a shift to warmer temperatures after 2.1 ka BP. Although such an inverse stratification by cooling of surface and warming of subsurface waters on the Labrador Shelf would suggest a diminished convection during the last two millennia compared to the mid-Holocene, it remains difficult to assess whether hydrographic conditions in the LC have had a significant impact on Labrador Sea deep-water formation.



## 1 Introduction

Since the early Holocene, the Labrador Sea is an important region of the global thermohaline circulation system through the formation of intermediate LSW in the central basin (Clarke and Gascard, 1983; Myers, 2005; Rhein et al., 2011; Yashayaev, 2007), which forms the upper part of North Atlantic Deep Water (NADW) and thereby contributes to the strength of the

Atlantic Meridional Overturning Circulation (AMOC). Modern type winter convection in the Labrador Sea (e.g. Hillaire-Marcel et al., 2001; Hoogakker et al., 2015) was likely established through a combination of diminished meltwater run-off following the final deglaciation of the LIS at about 7 ka BP (e.g. Jennings et al., 2015; Ullman et al., 2016) and the early Holocene strengthening of the WGC bringing a larger proportion of warm, more saline Atlantic waters into the Labrador Sea (e.g. Lloyd et al., 2005, Lochte et al., 2019, Seidenkrantz et al., 2013; Sheldon et al., 2016). In turn, high surface water

buoyancy may have been a limiting factor in LSW formation during the early Holocene (e.g. Hillaire-Marcel et al., 2001; Hoogakker et al., 2015). Today, the rate of Labrador Sea convection is controlled by two main processes: heat loss to the atmosphere and the relative supply of buoyant freshwater to the convection region. While changes in the heat loss are predominantly related to atmospheric forcing, Labrador Sea salinity is controlled mainly by two freshwater supply routes (Wang et al., 2018): the eastern route via the EGC that mixes with the Irminger Current (IC) south of Greenland to form the

WGC, and the western route via the LC, the extension of the Baffin Current (BC; Fig. 1). While early studies suggested the LC as the main source of central Labrador Sea freshwater (Lazier, 1973, 1988; Khatiwala et al., 1999), more recent work advocated the WGC as the main supplier of freshwater (Cuny et al., 2002; Straneo, 2006; Schmidt and Send, 2007).

In order to assess the relative roles of the WGC and LC in regulating the freshwater budget and convection of the Labrador

Sea, paleoceanographic reconstructions of these two main currents are critical. While several studies have provided middle to late Holocene reconstructions of the northward flowing WGC along the western coast of Greenland (Krawczyk et al., 2010; Lloyd et al., 2007; Moros et al., 2016; Møller et al., 2006; Seidenkrantz et al., 2007; Sha et al., 2017) or the IC (Moffa-Sánchez et al., 2014; Moffa-Sánchez and Hall, 2017), there is less evidence for water mass conditions in the southward flowing LC, as paleoceanographic reconstructions from the Labrador Shelf have mainly focussed on late deglacial to early

Holocene meltwater run-off from the LIS (e.g. Jennings et al., 2015, Hoffman et al., 2012; Lewis et al., 2012; Hillaire-Marcel et al., 2007) and its impact on LC strength (Rashid et al., 2017). Conversely, most reconstructions of sea surface temperature and sea ice cover focus on the region around Newfoundland (Keigwin et al., 2005; Solignac et al., 2011; Sicre et al., 2014, Sheldon et al., 2015, 2016) and Orphan Knoll (Hoogakker et al., 2011; 2015), whereas very little information is available about middle to late Holocene water mass conditions in the central LC, as high-resolution late Holocene sediment

records from the Labrador Shelf and Slope are sparse.

Here, we provide detailed reconstructions of LC temperature and salinity changes since the middle Holocene, in order to discuss the currents potential role in central Labrador Sea convection. Paleoceanographic reconstructions are based on a





high-resolution sediment record from the southern Labrador Shelf, providing information about sea surface and bottom water temperatures, freshening and sea ice cover of Labrador Shelf waters during the last 6,000 years.

**2 Oceanographic Setting**

The Labrador Sea is the coldest and freshest basin of the North Atlantic (Yashayaev, 2007) and is one of the major areas of

open ocean convection (Marshall and Schott, 1999). It plays a crucial role in influencing the strength and variability of the AMOC through the formation of LSW, which builds the upper part of NADW (Böning et al., 2006; Schmidt and Send, 2007; Schmitz and McCartney, 1993, Yashayaev, 2007; Yashayaev and Loder, 2009; 2016). Intermediate LSW forms through convection following an intense cooling of central Labrador Sea surface waters during particularly cold winters in North America (Clarke and Gascard, 1983). The annual variability of LSW formation was found to correlate with the NAO, the

dominant mode of atmospheric variability in the North Atlantic region (Hurrell, 1995). The mode of the NAO wields a major influence on temperature and precipitation patterns across the North Atlantic region and is defined by the difference in atmospheric pressure at sea level between the Icelandic low and the Azores high pressure systems (Hurrell, 1995). Positive NAO phases (NAO$^{+}$) are generally associated with strong north-westerly winds. These strong winds bring Arctic air southward, promoting enhanced winter cooling in the Labrador Sea region and thus stimulating deep convection (Dickson et

al., 1996) and fostering the southward advection of sea ice (Dickson et al., 2000; Deser et al., 2000, Drinkwater, 1996).

The Labrador Sea receives freshwater through its bordering currents, the WGC and the LC (Wang et al., 2018). As an extension of the BC, the LC flows south-eastwards along the continental margin of Labrador and Newfoundland and exports buoyant freshwater from the Arctic Ocean to the subtropical North Atlantic (Fig. 1a). The LC is divided into two branches, a

shallow inner branch that flows over the Labrador Shelf, and an outer branch that is centred along the upper continental slope (Lazier and Wright, 1993). The inner LC is the immediate continuation of the BC (Cuny et al., 2002), augmented by Arctic Water outflow from Hudson Bay (Straneo and Saucier, 2008), while the outer LC receives a significant proportion of Atlantic waters by the westward retroflection of the West Greenland Current (WGC), which itself is formed through the cold East Greenland Current (EGC), and the warm, saline IC (Cuny et al., 2002). Mixing of the inner and outer LC branches

occurs in the saddles between the prominent banks on the Labrador Shelf (Vilks, 1980; see Fig. 2). Just south of Cartwright Saddle, in the vicinity of Flemish Cap and Newfoundland, the LC flows into the subpolar North Atlantic (Fig. 1).



### 3 Material and Methods

#### 3.1 Material

Gravity core MSM45-31-1 (1150 cm core length) was recovered from the Labrador Shelf at 54°24.74 N, 56°00.53 W, at 566 m water depth (Fig. 1a) during the R/V Maria S. Merian cruise MSM45 in August 2015 (Schneider et al., 2016). The core
site is located next to the Cartwright Saddle, off Hamilton Inlet, in a depression associated with an over 600 m deep basin that is part of the Labrador Marginal Trough along the inner Labrador Shelf. The sediment column consists of homogenous olive grey, silty, clayey mud and was subsampled in 5-cm intervals on board the research vessel. A CTD (Conductivity-Temperature-Depth) deployment at nearby station 30 at 54°28.53N, 56°04.33W provides depth profiles of temperature and salinity (Fig. 1b) of southern Labrador Shelf waters at the time of collection. A downward increase in temperature and
salinity below 200 m depth shows that the inner LC overlies warmer and more saline waters of the outer LC (Fig. 1b, 2).

#### 3.2 AMS radiocarbon dating

The stratigraphy of MSM45-31-1 is based on 12 Accelerator Mass Spectrometry (AMS) [14]C measurements on mixed calcareous benthic foraminifera at the Leibniz Laboratory of Kiel University (CAU), Germany. Each sample contained over 1000 specimens (about 5 mg), which were picked from 1 cm thick sediment slices in 1-m intervals down-core. The [14]C dates
were calibrated using Calib 7.1 (Stuiver and Reimer, 1993; Stuiver et al., 2017) and the Marine13 dataset (Reimer et al., 2013) with a reservoir correction ($\Delta R$) of 144 ± 38 years, based on the present Labrador Shelf marine radiocarbon correction (McNeely et al., 2006). Dates are reported in calibrated years BP (before present) and are presented in Table 1.

#### 3.3 Alkenone biomarkers

218 bulk sediment samples were analysed in 5-cm intervals at the Biomarker Laboratory, Institute of Geosciences, Kiel
20  University. Long-chained alkenones ($C_{37}$) were extracted from 2–3g homogenized bulk sediment, using an Accelerated Solvent Extractor (Dionex ASE-200) with a mixture of 9:1 (v/v) of dichloromethane:methanol (DCM:MeOH) at 100°C and 100 bar $N_2$ (g) pressure for 20 min. Extracts were cooled at -20°C and brought to near dryness by Syncore polyvap at 40°C and 490 mbar. For the identification and quantification of $C_{37:2}$, $C_{37:3}$ and $C_{37:4}$, we used a multi-dimensional, double column gas chromatography (MD-GC) set up with two Agilent 6890 gas chromatographs (Etourneau et al., 2010). The addition of an
25  internal standard prior to extraction (cholestane [$C_{27}H_{48}$] and hexatriacontane [$C_{36}H_{74}$]) allowed quantification of the organic compounds that are reported in nanograms per gram dry bulk sediment. The proportion of each unsaturated ketone was obtained by integration of peak areas of the different compounds in the respective chromatograms. Alkenone concentrations (sum of $C_{37:2}$, $C_{37:3}$ and $C_{37:4}$) are used as indication of alkenone productivity.



### 3.3.1 Sea Surface Temperature estimates

Sea surface temperature estimates are based on the $U^K_{37}$ index (Brassell et al., 1986) and the $U^{K'}_{37}$ index (Prahl and Wakeham, 1987). The $U^K_{37}$ index was calculated according to Brassell et al. (1986): $U^K_{37} = (C_{37:2} - C_{37:4}) / (C_{37:2} + C_{37:3} + C_{37:4})$, while the $U^{K'}_{37}$ index was calculated using the calibration of Prahl and Wakeham (1987): $U^{K'}_{37} = (C_{37:2}) / (C_{37:2} + C_{37:3})$, both resulting in standard deviations of 0.01. Rosell-Melé (1998) and Bendle and Rosell-Melé (2004) point out that $U^K_{37}$ based SST estimates down to 6°C are more congruent with modern observations than those based on $U^{K'}_{37}$. However, $U^K_{37}$ based SST estimates could be biased with respect to absolute values at $C_{37:4}$ values above 5% (Rosell-Melé, 1998), which is the case here, and should therefore be interpreted with caution.

### 3.3.2 %$C_{37:4}$ – proxy for meltwater or sea ice cover?

Several studies have suggested an increase in the production of tetra-unsaturated $C_{37}$ ($C_{37:4}$) in polar and subpolar surface waters at lower salinities (Bendle et al., 2005; Blanz et al., 2005; Rosell-Melé, 1998; Rosell-Melé et al., 2002; Sicre et al., 2002). Thus, the proportion of $C_{37:4}$ relative to the sum of alkenones [%$C_{37:4} = C_{37:4} / (C_{37:2} + C_{37:3} + C_{37:4})$] in sedimentological records from the North Atlantic is often used to qualitatively indicate changes in salinity. However, several studies have observed no, or even an inverse relationship between %$C_{37:4}$ and salinity (Schwab and Sachs, 2011; Sikes and Sicre, 2002; Filippova et al., 2016; Theroux et al., 2010; Toney et al., 2010). Bendle et al. (2005) found that higher proportions of $C_{37:4}$ are produced in low productivity polar waters, probably because $C_{37:4}$ proportions vary in different alkenone-producing haptophyte species (Conte et al., 1995; Marlowe et al., 1984). A culture experiment by Chivall et al. (2014) showed that certain, not coccolith-bearing, prymnesiophytes like *Chrysotila lamellosa* produce between 30 – 44% $C_{37:4}$ in a positive relationship to salinity. Nonetheless, according to Bendle et al. (2005), highest $C_{37:4}$ concentrations in surface sediments from the polar Atlantic delineate the mean position of the sea ice margin off Greenland. In addition, preliminary core-top studies imply that high (>15%) abundances of $C_{37:4}$ are restricted to surface sediments within the limits of seasonal sea ice extent in Baffin Bay and the Labrador Sea (Thomas Blanz, unpublished data). Although the exact environmental factors that affect the production of $C_{37:4}$ remain ambiguous, we interpret higher proportions of $C_{37:4}$ (>15%) as an indicator for sea ice margin conditions.

### 3.4 Mg/Ca measurements

For Mg/Ca measurements, 20 – 70 specimens of *Nonionellina labradorica* were handpicked from the >315 µm size fraction, weighed, and crushed between two glass plates. Two thirds of the crushed sample were transferred into pre-leached Eppendorf vials and cleaned following the full protocol of Martin and Lea (2002), including a reductive and oxidative cleaning step and a final leaching step with 0.001 N HNO₃. After dissolving and diluting the samples in 0.1 N HNO₃, they were measured with an ICP-OES instrument with radial plasma observation at the Institute of Geoscience, Kiel University. The analytical error for Mg/Ca analyses was 0.1% relative standard deviation. The JCP-1 standard material (Hathorne et al.,

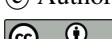



2013) was measured each $6^{th}$ sample for accuracy and drift correction. Additional accuracy control was applied by measurements of standard BAM RS3 (Greaves et al., 2008) each $60^{th}$ sample. Trace elements (Fe, Al and Mn) were monitored to exclude possible contaminated or coated samples from the dataset. Mg/Ca values exceeding 2.4 mmol/mol were considered unreliable and are marked as flyers in the dataset (Fig. 4d). Based on seven duplicate downcore sample

measurements, we obtained an analytic precision of 0.2 mmol/mol Mg/Ca, which translates into 2°C in respective temperature estimates. Bottom water temperature (BWT) reconstructions are based on the calibration of Skirbekk et al. (2016) for the subpolar North Atlantic region. As the calibration is based on a temperature range of 1 – 4°C, estimates exceeding 4°C may be less accurate, which is the case for about 35% of the samples reported here.

### 3.5 Stable isotope measurements

For stable isotope measurements, the remaining one third of the crushed samples of *N. labradorica* were cleaned with ethanol absolute, decanted and dried at 40°C. Stable isotope analyses were carried out at the Leibniz Laboratory for Radiometric Dating and Stable Isotope Research in Kiel. A Finnigan MAT 253 mass spectrometer coupled with a Kiel IV carbonate preparation device was used. Results were calibrated to the Vienna Pee Dee Belemnite (V-PDB) scale. Based on 11 downcore duplicate measurements, the standard deviation is 0.11 ‰. To estimate the $\delta^{18}O_w$ (seawater)*,* the value of $\delta^{18}O_c$

(‰ V-PDB) was first corrected for vital effects of *N. labradorica* by subtracting 0.15 ‰ (Rasmussen and Thomsen, 2009). The corrected $\delta^{18}O_c$ values were then applied in the Shackleton et al. (1974) equation together with the Mg/Ca-derived temperature estimates. To correct for the effect of global ice volume on the oceans $\delta^{18}O_w$ signature, a sea-level correction of 0.0083‰ per m (1‰ for 120 m of sea level), derived from the relative sea level curve by Austermann et al. (2013), was applied. Based on duplicates, the standard deviation of the calculated ice volume corrected $\delta^{18}O_{w-ivc}$ is 0.35 ‰.

## 4 Results

### 4.1 Chronology

The chronology of core MSM45-31-1 is based on linear interpolation of the calibrated median probability ages that are presented in Table 1. The age-depth model is presented in Fig. 3. The sedimentation rate varies between 0.11 and 0.35 cm per year and is lowest in the deeper part of the core, from 1150 – 900 cm depth (Fig. 3), probably due to compressional

effects resulting from gravity coring. The chronological resolution is between 3 – 9 years per cm, so the core provides a sub-decadal record of the late Holocene.

### 4.2 Temporal variability in southern Labrador Shelf waters

Alkenone concentrations range from 20 – 80 ng/g and, despite major fluctuations, gently increase on average towards the top of the core (Fig. 4a). The $U^{K'}_{37}$ index ranges from 0.3 to 0.43 and respective SST estimates range from 8 °C to 12 °C (Fig.

4b). The $U^K_{37}$ index and SST estimates range from 0.05 to 0.25 and from 0 °C to 6 °C, respectively (Fig. 4c). The proportion





of $C_{37:4}$ ranges from 8 % to 22 % and shows lowest values between 5.6 ka BP and 2.1 ka BP (Fig. 4d). Mg/Ca ratios range from 1.3 mmol/mol to 3 mmol/mol (excluding values exceeding 2.4 mmol/mol, see 3.4 above) (Fig. 4e). BWT estimates range from 0.6°C to 9.9°C and indicate a shift to warmer temperatures at 2.1 ka BP (Fig. 4e). The $\delta^{18}O_c$ (‰-VPD) of *N. labradorica* ranges from 3.2‰ to 4.0‰ and does not show any significant minima/maxima (Fig. 4f). The temperature and

ice volume corrected $\delta^{18}O_{w-ivc}$ (‰-SMOW) record ranges from –0.3‰ to 2.9‰ and shows a small shift to more positive values after 2.1 ka BP (Fig. 4g). According to major changes in surface and bottom water conditions, the record has been divided into three main environmental intervals: From 6.2 to 5.6 ka BP, from 5.6 to 2.1 ka BP, and from 2.1 ka BP to present (Fig. 4).

### 4.2.1 From 6.2 to 5.6 ka BP (1150 – 1085 cm)

Between 6.2 – 5.6 ka BP, the alkenone concentration is relatively low with average values of 33 ng/g. The $U^{K'}_{37}$ and $U^{K}_{37}$ indices and relative SST estimates show the lowest values of the record, suggesting a period of cool surface waters on the Labrador Shelf, while the proportion of $C_{37:4}$ is high at about 18 %. Mg/Ca levels and BWT estimates are also relatively low at 1.6 mmol/mol and about 3°C, respectively. The $\delta^{18}O_c$ is at about 3.5 ‰ and the $\delta^{18}O_{w-ivc}$ is at about 0 ‰.

### 4.2.2 From 5.6 to 2.1 ka BP (1080 – 435 cm)

Between 5.6 – 2.1 ka BP, alkenone concentrations vary between 25 ng/g and 80 ng/g. Two peaks up to 60 ng/g and 55 ng/g at 5.4 ka BP and 3.8 – 3.0 ka BP are evident (Fig. 4a). $U^{K'}_{37}$ indices are in medium, while $U^{K}_{37}$ indices are in maximum ranges in this interval, leading to SST estimates of 8 – 10 °C and 2 – 6 °C, respectively. $U^{K}_{37}$ based SST estimates show two peaks of 5.8 °C and 6 °C at 5.4 ka BP and 3.8 – 3.0 ka BP. The proportion of $C_{37:4}$ shows the lowest values in this interval with an average of 12 % and two minima of 9 % and 8 % at 5.4 ka BP and 3.1 ka BP. Despite the warming of surface waters,

BWT reconstructions remain at average values of 3.1 °C, which corresponds to modern conditions at the core site (Fig. 1b, 2). However, the BWT record shows two warm peaks of 7.7 °C and 9.8 °C at 5.4 ka BP and 5.2 ka BP, respectively, and a third warm peak to 7 °C at 3.8 ka BP. The $\delta^{18}O_c$ record shows a minor long-term increase with average values of 3.6 ‰. The $\delta^{18}O_{w-ivc}$ record shows an increasing trend towards positive values of about 0.5 ‰, except for a maximum of about 2 ‰ at 5.2 ka BP, suggesting a short increase in bottom water salinity.

### 4.2.3 From 2.1 ka BP to present (435 – 0 cm)

The interval after 2.1 ka BP shows an increase in alkenone concentration that peak to 75 ng/g at 0.7 ka BP, with average values of 48 ng/g. The $U^{K'}_{37}$ based SST record shows higher fluctuation in this interval with temperatures between 8.5 °C and 11.5 °C, while the $U^{K}_{37}$ based SST record reveals a general decrease with fluctuations between 0 °C and 6 °C and average values of 2.2 °C. The $C_{37:4}$ record is characterized by a shift to values of 17 % on average, suggesting an increase in

sea ice after 2.1 ka BP. Especially in this interval, the use of either $U^{K}_{37}$ or $U^{K'}_{37}$, results in opposite SST trends. Nonetheless, the use of the $U^{K}_{37}$ index results in SST estimates closer to the range of modern values in the upper water column (Fig. 2)





than those provided by the $U^{K'}_{37}$ index. The BWT reconstructions display higher values after 2.1 ka BP, with average temperatures of 6 °C. The $\delta^{18}O_c$ record continues with a slight increase to more positive levels, while the $\delta^{18}O_{w\text{-}ivc}$ record shows higher fluctuations and generally more positive values in this interval as well as a peak to nearly 3 ‰, in correspondence with the bottom water warm peaks.

## 5 5 Discussion

### 5.1 The reliability of sea surface and bottom water temperature estimates with regard to modern conditions

$U^K_{37}$ and $U^{K'}_{37}$ based SST estimates differ substantially in this region, due to relatively large proportions of $\%C_{37:4}$. While $U^{K'}_{37}$ based SST estimates with an average of 9 °C appear too warm for this region, $U^K_{37}$ based SST estimates with an average of 3 °C are congruent with modern observations of SSTs at the core site. Although the surface temperature measured in August 2015 was 6 °C (Fig. 1b), the annual mean sea surface temperature at the core site is below 2 °C (Fig. 2). However, alkenones neither reflect peak summer (i.e. August) nor annual mean temperatures, but rather record the average temperatures of the plankton blooming season, which on the Labrador Shelf, today, is between May and November (Frajka-Williams and Rhines, 2010; Harrison et al., 2013). Thus, at our core site, average $U^K_{37}$ based SST estimates of our down-core record (3.3 °C) as well as of the most recent core top sample (3.3 °C) agree very well with the modern average sea surface temperature between May and November of 3.5 °C (WOA13; 1955 – 2012; Locarnini et al., 2013). Therefore, we assume that the $U^K_{37}$ index provides the most accurate SST estimates for our record, although we are aware that alkenone-based temperature estimates become less robust with increasing proportions of $\%C_{37:4}$ and need to be interpreted with caution (Rosell-Melé, 1998).

The down-core BWT mean value of 3.8 °C based on benthic foraminiferal Mg/Ca ratios is slightly above the average $U^K_{37}$-based SST estimates in our record (3.3 °C; Fig. 5), but consistent with annual mean temperatures above 3.5 °C seen today at about 500 m depth. Although such a water column temperature inversion appears counterintuitive, it is likely caused by inflowing warmer and more saline subsurface waters (Fig. 1b) of the outer branch of the LC that receives a large proportion of the westward retroflection of the WGC, which, in turn, is a mixture of the cold EGC and the warm, saline IC. The outer LC, fed by the WGC, reaches the deeper regions on the Labrador Shelf, while the upper 200 m of the water column are dominated by the cold and shallow inner LC, which carries a higher proportion of Arctic waters from Baffin Bay. Thus, we interpret the $U^K_{37}$-based SST estimates to reflect the inner LC, while Mg/Ca-based BWT estimates rather reflect conditions of the outer LC and the westward retroflection of the WGC.

### 5.2 Middle to late Holocene phases of Labrador Shelf oceanography

Based on the surface and bottom water records presented here, we differentiate three main climatic and oceanographic intervals in Labrador Shelf waters during the middle to late Holocene. From 6.2 to 5.6 ka BP, the LC experienced a cool



period with a strong sea ice cover that is also evident on the north-eastern side of the Labrador Sea (Disko Bugt; e.g. Moros et al., 2016). This cold interval was followed by generally warmer conditions in Labrador Shelf surface waters between 5.6 and 2.1 ka BP associated with the Holocene Thermal Maximum in the western Arctic (Kaufman et al., 2004). In addition to the gradual cooling trend of the last 3,000 years, our $C_{37:4}$ record implies a recurrence of stronger sea ice cover after 2.1 ka

BP. Simultaneously, a shift to warmer bottom water temperatures after 2.1 ka BP likely corresponded to an increased supply of warmer Atlantic waters with the westward retroflection of the WGC reaching deeper regions on the Labrador Shelf.

**5.2.1 Cold Labrador Current (6.2 – 5.6 ka BP)**

From 6.2 to 5.6 ka BP, low sea surface and bottom water temperatures suggest that both surface and subsurface waters were dominated by cold, Arctic water masses from Baffin Bay, with a significant sea ice cover indicated by relatively high %$C_{37:4}$

(Fig. 6e,f,h). We associate this period with a cold inner LC caused by strong advection of Arctic waters and sea ice from Baffin Bay. Colder conditions with a longer annual sea ice cover have also been detected in a northern Labrador fjord between 7 and 5.8 ka BP (Richerol et al., 2016). A similar scenario is also seen south of our core in Trinity Bay, Newfoundland, between 7.2 and 5.5 ka BP (Sheldon et al., 2015), with a period of cold surface waters and seasonal sea ice cover, most likely sourced from the Arctic via the LC. Moreover, Solignac et al. (2011) estimated cold SSTs around

Newfoundland during the Mid-Holocene and ascribed these conditions to a stronger than present LC, which may have been caused by accelerated atmospheric (westerly wind) circulation patterns (Jessen et al., 2011) combined with strong melting in the Arctic. In contrast to this scenario, however, a decline in sortable silt (SS) mean size on the northern Labrador Shelf has been interpreted to reflect a relatively weak LC during this period, (Site Hu2006040-40, Rashid et al., 2017, Fig. 6c).

On the north-eastern rim of the Labrador Sea, in Disko Bugt (Greenland), high amounts of %$C_{37:4}$ were also evident around 6

20   ka BP (Moros et al., 2016; Fig. 6b), which are in agreement with our findings. The authors interpreted these high proportions of %$C_{37:4}$ to indicate strong meltwater run-off. At the same time, between 6.2 and 5.5 ka BP, they identified an increase in sea-ice associated diatoms and an abrupt sub-surface cooling (Moros et al., 2016), suggesting that the high %$C_{37:4}$ could also be interpreted as stronger sea ice cover. The conditions in Disko Bugt may well have been related to a cold WGC associated with a cooling in the EGC at about this time (Perner et al., 2015; Müller et al., 2012; Ran et al., 2006). A distinct cooling

episode between 7 and 5 ka BP is also seen in Greenland ice core data (O'Brien et al., 1995), which was likely related to changes in atmospheric circulation leading to increased storminess in the NE United States (Noren et al., 2002). A cold spell is evident in the Camp Century ice core record as well, with a pronounced decrease of $\delta^{18}$O values at 5.8 – 5.6 ka BP (Vinther et al., 2009), consistent with a temporary cooling-freshening in oceanic conditions affecting northeast Baffin Bay. Additionally, a marked temperature drop in the northern North Atlantic (e.g. Moros et al., 2004; Telesiński et al., 2014) is

likely linked to the most pronounced North Atlantic Holocene ice rafted debris event (Bond et al., 2001). The coincidence of cool periods in records from the western, southwestern and eastern Labrador Sea, as well as in many other North Atlantic records (e.g., Bond et al., 2001; Moros et al., 2004; Telesiński et al., 2014) suggests that it was a widespread phenomenon in the western North Atlantic region.





### 5.2.2 Sudden warming followed by gradual cooling of the Labrador Current (5.6 – 2.1 ka BP)

The start of this interval at 5.6 ka BP is marked by a pronounced increase in SSTs to over 6°C and a decline of %$C_{37:4}$ to about 10% (Fig. 6e+f), implying a substantial warming of Labrador Shelf surface waters and less sea ice cover. Two warm peaks can also be seen in bottom waters at the beginning of this interval, coeval with SST maxima (Fig. 6e,h). Warmer

conditions on the Labrador Shelf may be related to the shallowing of the Arctic channels at about 6 ka BP, which led to a decrease in Arctic water flow to Baffin Bay and instead enhanced the export through Fram and Denmark straits along the East Greenland coast (Williams et al., 1995). Warmer conditions were also indicated by the benthic foraminiferal assemblage of Trinity Bay, Newfoundland, that showed increases in fresh food supply and primary productivity between 5.7 and 4.85 ka BP (Sheldon et al., 2015). Further south, off Cape Hatteras, Cléroux et al. (2012) described increased surface-

water salinities between 5.2 and 3.5 ka BP, which they attributed to decreased export of colder and fresh waters from the north, allowing a northward shift of the Gulf Stream. In contrast to this proposed weakening of the LC, Rashid et al. (2017) identified a short-lived increase in SS mean size in northern Labrador Shelf core Hu2006-040-040 (Fig. 6c), suggesting an intensification of the LC. This latter observation is in agreement with dinocyst assemblages of two Newfoundland records from Bonavista and Placentia Bay (Solignac et al. 2011) suggesting fresher and colder sea surface conditions between 5.6

and 4 ka BP, which the authors link to enhanced meltwater supply during a warm period in the Arctic until 4.5 – 4 ka BP.
In Disko Bugt, western Greenland, the interval from 5.5 – 3.5 ka BP is marked by a return to relatively warm subsurface conditions and a gradual decrease of %$C_{37:4}$ values following the peak at 6 ka BP (Moros et al., 2016; Fig. 6b), in agreement with our findings. Diatom and dinocyst assemblages both show relatively mild surface waters despite a gradual trend towards cooler conditions, which is also evident in our SST record between 5.5 and 3.5 ka BP. A reduction in *N. labradorica*

indicates lower surface water productivity that is also suggested by dinocyst assemblages (Ouellet-Bernier et al., 2014), which Moros et al. (2016) link to the increased strength and/or warmth of the WGC. The enhanced influence of the WGC may have resulted from a strong and relatively warm IC that is reported from the East Greenland shelf (Jennings et al., 2002, 2011) and southwest and south of Iceland (e.g. Knudsen et al., 2008). The strong influence of the WGC in Disko Bugt may also have been the result of a largely land-based ice sheet and reduced meltwater runoff from the GIS after 6 ka BP (Briner

et al., 2010; Weidick and Bennike, 2007; Weidick et al., 1990) with minimum ice extent from ca. 5 – 3 ka BP (Briner et al., 2014). This period as well corresponds to relatively warm air temperatures recorded in Greenland ice cores (e.g. Camp Century, Moros et al., 2016), which were likely a result of the Holocene Thermal Maximum between 5 and 3 ka BP over central to southern Greenland (Kaufman et al., 2004). Additionally, several lake records near Jakobshavn Isbræ display high productivity under relatively warm terrestrial conditions, and one of the lakes, North Lake (Axford et al., 2013), indicates

relatively high chironomid-based temperatures. High lake levels linked to warmer conditions are also reported in the Kangerlussuaq region, just south of Disko Bugt (Aebly and Fritz, 2009).



In our record, a second warm peak at 3 ka BP is evidenced by the lowest $\%C_{37:4}$ and the highest SSTs (Fig. 6e+f). The warm temperatures around 3 ka BP correlate with a period of negative NAO conditions (Olsen et al., 2012; Fig. 6a), which generally result in weaker north-westerly winds and reduced transport of Arctic waters through the Canadian gateways, causing a weaker LC, milder winters and shorter sea ice seasons. Warmer sea surface conditions are also seen in Placentia
Bay, Newfoundland, indicated by the lowest numbers of sea-ice indicator species *I. minutum* between 3.2 and 2.2 ka BP (Solignac et al., 2011). These warmer conditions around Newfoundland correlate with the lowest $\%C_{37:4}$ in our record, suggesting that both regions experienced a reduction in sea ice cover, possibly caused by generally weak north-westerly winds. In agreement with our records, warmer annual mean temperatures were also detected in Greenland ice core GISP2 between 4 and 2 ka BP (Alley et al., 1999), potentially also related to the predominantly negative NAO conditions at that
time. In contrast to this warm episode with likely low sea ice export from the Arctic region, northern Labrador Shelf core Hu2006-040-040 has been interpreted to imply a relatively strong current at this time (Rashid et al., 2017; Fig. 6c). Although the strength of the LC has generally been tied to meltwater drainage and enhanced freshwater fluxes during the early Holocene, no such linkage has been found for the middle to late Holocene (Rashid et al., 2017). Thus, the role of sea ice and freshwater export on the strength of the LC remains uncertain.

After about 3 ka BP and coeval with the Neoglaciation (Kaufman et al., 2004; Vinther et al., 2009), LC surface waters cooled gradually. The onset of the Neoglaciation between 3.2 – 2.7 ka BP was associated with a cooling in the Arctic that reduced meltwater and sea ice export (Scott and Collins, 1996) and/or iceberg production (Andresen et al., 2011). Correspondingly, reduced meltwater and cold bottom water conditions were detected in Disko Bugt (Greenland) between 3.5
and 2 ka BP (Lloyd et al., 2007). A similar cooling trend was implied by a general decrease in Atlantic water species in Disko Bugt after 3.5 ka BP (Perner et al., 2011). Lake deposits from southern Greenland indicate dry and cold conditions from 3.7 ka BP (Andresen et al., 2004) and shortly after 3.0 ka BP (Kaplan et al., 2002), respectively, while Vinther et al. (2005) recognized a sharp decrease in the amplitude of the GRIP annual $^{18}$O cycles, suggesting a cooling of 1 – 2 °C ca. 3000 years ago. On the north Iceland shelf, temperatures dropped significantly at ca. 3.2 ka BP (Eiríksson et al., 2000; Jiang
et al., 2002), and in the North Atlantic, Bond et al. (1997) identified a strong ice rafted debris event at 2.8 ka BP. Hence, the gradual cooling seen in our surface temperature record after 3 ka BP is in agreement with many North Atlantic records that showed colder conditions associated with the Neoglaciation.

### 5.2.3 Subsurface warm water intrusions on the Labrador Shelf despite further surface cooling (2.1 ka BP – present)

After 2.1 ka BP, the SST record shows a continuation of the gradual cooling trend that started around 3 ka BP with the onset
of the Neoglaciation. This continued cooling corresponds to the shift from predominantly negative to predominantly positive NAO conditions (Olsen et al., 2012; Fig. 6a), which would have resulted in generally stronger north-westerly winds and colder winters in the Labrador Sea region. The colder surface waters with stronger sea ice cover evident in our record during the last 2,100 years suggest that the inner LC received a strong supply of Arctic waters from Baffin Bay in response to the



Neoglacial cooling and generally positive NAO conditions. In support of our findings, colder conditions and an increase in the sea ice duration were also recorded in Trinity Bay, Newfoundland, after 2.1 ka BP (Sheldon et al., 2015). Additionally, a gradual surface water cooling of about 2°C during the last 2000 years was also evident in another core from Newfoundland (AI07-12G; Sicre et al., 2014; Fig. 6d), which further supports the surface water cooling trend observed in our record.

Coeval with this Neoglacial cooling trend, northern Labrador Shelf core Hu2006-040-040 shows a gradual decrease in SS mean size after about 3 ka BP (Fig. 6c), implying a LC weakening. The general positive correlation of our SST record with the SS mean size record in Hu2006-040-040 implies that colder LC surface temperatures correspond to a weaker current. Although LC vigour was tied to early Holocene freshwater discharge, no such linkage is apparent during the middle to late Holocene (Rashid et al., 2017). The authors, however, suggested that the LC weakening of the last 3,000 years was related to

the Neoglacial cooling and enhanced sea ice production, which would have locked up larger amounts of freshwater, resulting in diminished supply of freshwater and a weakening of the LC. The negative correlation of our %$C_{37:4}$ record and the SS record of core Hu2006-040-040 does support this scenario of enhanced sea ice cover reducing LC strength. Another explanation for the LC weakening and cooling during the last 3,000 years may be that the shift to predominantly positive NAO-like conditions (Olsen et al., 2012; Fig. 6a) caused stronger north-westerly winds that promoted the southward

advection of sea ice (Drinkwater, 1996) while also causing a cooling of Labrador Sea surface waters. This would, in turn, lead to a reduced density of offshore waters relative to inshore waters, which would decrease the baroclinic pressure gradient and weaken LC transport (Dickson et al., 1996).

Despite the gradual cooling trend of the LC during the last 3,000 years (Fig. 6e), bottom waters show a shift to warmer

conditions after 2.1 ka BP (Fig. 6h). The shift to higher BWTs occurred simultaneously to the increase in %$C_{37:4}$ (Fig. 6f), implying a strongly stratified water column that was influenced by two different water masses. This scenario can be compared to modern conditions of warmer waters underlying cold waters at our core site (Fig. 1, 2). Today, the cold inner LC only dominates Labrador Shelf waters down to about 200 m depth (Lazier and Wright, 1993), while the deeper part of the shelf, such as our core site at 566 m depth, is influenced by the warmer outer LC that is largely supplied by the westward

retroflection of the WGC. Hence, we ascribe the warmer conditions seen in Labrador Shelf bottom waters after 2.1 ka BP to a strengthening or warming of the WGC carrying warmer Atlantic waters to the deeper regions on the Labrador Shelf. An increased flux of Atlantic sourced water from the IC was also seen in the central Labrador Sea, indicated by decreased abundances of polar water planktic foraminifera species *N. pachiderma* sinistral (%Nps) at about 2.3 – 2.1 and 1.6 ka BP (RAPiD-35-COM; Moffa-Sánchez and Hall, 2017; Fig. 6i), which seem to correspond to the subsurface warming peaks in

our record. Despite the generally warmer conditions in our BWT record during the last 2,100 years, the record displays a trend returning to colder temperatures in this interval. Such a cooling trend is also evident in core RAPiD-35-COM and has been linked to a weakening in LSW formation (Moffa-Sánchez and Hall, 2017).



While our surface records imply the formation of a cold inner LC with a larger seasonal sea ice cover during the last 2,100 years, central Labrador Sea records document enhanced inflow of the warm and saline IC (Moffa-Sánchez and Hall, 2017) that is reflected by subsurface warming and higher salinity on the Labrador Shelf (Fig. 4). According to Gelderloos et al. (2012), cold and fresh surface waters are thought to inhibit convection in the Labrador Sea, while modern observations of

warming and increased salinity in central Labrador Sea subsurface waters correlate with episodes of weak Labrador Sea convection (Yashayaev, 2007). Furthermore, extending observational data by proxy records for the last 1,200 years, Thornalley et al. (2018) report a strong reduction in LSW formation and DWBC flow during the last 150 years that seem to correspond to a trend to warmer and saltier subsurface waters in the Labrador Sea (Moffa-Sánchez et al., 2014). According to these studies, the reconstructed surface cooling and subsurface warming and salinity increase on the Labrador Shelf

starting at 2.1 ka BP would strongly suggest that LSW formation in the western Labrador Sea was reduced during the last two millennia compared to the mid-Holocene. In contrast, Moffa-Sánchez and Hall (2017) infer the opposite scenario for the last 2,000 years with a stronger SPG and intensified LSW formation during warmer eastern Labrador Sea conditions (see Fig. 6i). The latter would be congruent with the observations during more positive NAO conditions resulting in stronger convection (e.g. Zantopp et al., 2017).

Despite the ambiguous evidence of the coupling between LC temperature and LSW formation, it appears that a colder inner LC was coupled with a warmer and saltier IC and WGC during the last 2,100 years (see also discussion in Moffa-Sánchez et al., 2019). This link can be explained by a reduced LC strength – as more freshwater would have been locked up in sea ice – resulting in diminished supply of cold, Arctic waters into the North Atlantic Current, which, in turn, would have led to a

warmer IC. Nonetheless, it is currently still difficult to assess whether hydrographic conditions in the LC have had a significant impact on deep-water formation in the central basin or just responded to general atmospheric conditions across the Labrador Sea region, which may have been the most important driver controlling LSW production during the late Holocene.

Interestingly, Marchitto and de Menocal (2003) reported a significant Little Ice Age (LIA) cooling in intermediate water depths along the path of the Deep Western Boundary Current (DWBC) south of Newfoundland, corresponding to the cooling seen in our subsurface water record from the Labrador Shelf. This correspondence might indicate a coupling between outer LC and DWBC temperature changes at centennial time scale with the LSW formation being the link between both water masses. However, overall short-term variability in DWBC temperatures during the last 4,000 years does not exactly match

our record. Therefore, to fully assess the interaction between LC variability and LSW formation, more high-resolution records of changes in LSW production from the western Labrador Sea are required.



## 6 Conclusion

Overall, middle to late Holocene conditions in Labrador Shelf waters display variability that can partly be related to atmospheric forcing, such as NAO-like conditions. Between 6.2 and 5.6 ka BP, our records imply a cold episode with strong sea ice cover on the Labrador Shelf that has also been evident in other cores from the North Atlantic region. This cold

episode was followed by a generally warmer interval from about 5.6 to 2.1 ka BP corresponding to a late Holocene Thermal Maximum, interrupted by colder surface temperatures between about 4.5 and 3.5 ka BP. After about 3 ka BP, our SST record shows a gradual cooling trend in association with the Neoglaciation, which has also been observed in other surface records from the Labrador Sea region. However, at 2.1 ka BP our record shows an abrupt shift to enhanced sea ice cover and warmer bottom waters on the Labrador Shelf that corresponds to a shift from predominantly negative to predominantly positive

NAO-like conditions. We associate the cooling seen in surface waters with stronger north-westerly winds and harsher winters in the region during positive NAO, while the warming in bottom waters was possibly related to a stronger inflow of the westward retroflection of the WGC in response to a stronger supply of the IC that was seen in the central Labrador Sea. Our record implies that phases of enhanced sea ice cover on the Labrador Shelf corresponded to reduced LC strength during the last 6,000 years, which may be have been related to atmospheric conditions controlling the length of the sea ice season as

well as regulating the baroclinic pressure gradient that drives LC velocities. During the last 2,100 years, a reduced current and diminished freshwater supply via the LC – as more freshwater would have been locked up in sea ice – would have led to a diminished supply of freshwater through the western route to the SPG, which, in turn, could have resulted in saltier and denser NAC and IC (Fig. 1). At the same time, intense cooling of dense SPG surface waters during generally positive NAO conditions would have promoted winter convection and strengthened LSW formation in the central basin. Therefore, it

appears that positive NAO conditions may have been responsible for both increased Labrador Shelf sea ice cover during the last 2,100 years and reduced freshwater supply and LC strength, while enhancing winter cooling and convection in the central Labrador Sea. The potential indirect effect of enhanced sea ice cover and limited freshwater supply via the LC may thus have played an additional role in enabling deeper convection forming colder and saltier LSW, a positive feedback mechanism during positive NAO conditions.

## 7 Data availability

The data reported in this paper are archived in PANGAEA and available at https://doi.org/10.1594/PANGAEA.904693.

## 8 Author contribution

RS, JR and MK guided sediment and water column sampling at sea on the RV *Maria S. Merian*. AAL performed the Mg/Ca and stable isotope sample preparation and established the age model. TB performed the biomarker sample preparations and

measurements. DGS led the ICP-OES measurements. NA performed the stable isotope measurements. AAL led the data



compilation, interpretation, and writing of the manuscript in discussions with RS, JR and MK. All co-authors contributed to improve the manuscript.

**9 Competing interests**

The authors declare that they have no conflict of interest.

5  **10 Acknowledgements**

We wish to thank the captain and crew of the RV *Maria S. Merian* as well as the scientific team for their great help during the MSM45 cruise. We thank Karen Bremer and Silvia Koch for technical assistance with the ICP-OES and biomarker analyses, respectively. This study was supported by a PhD fellowship to AAL through the Helmholtz Research School on Ocean System Science and Technology (www.hosst.org) at GEOMAR Helmholtz Centre for Ocean Research Kiel (VH-KO-
10  601) and Kiel University.

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

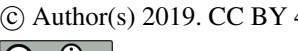



**Table 1: Twelve AMS radiocarbon dates, calibrated in Calib7.1 (Stuiver and Reimer, 1993; Stuiver et al., 2017) using the Marine13 dataset (Reimer et al., 2013) with a reservoir correction of ΔR = 144 ± 38 years (McNeely et al., 2006).**

| AMS laboratory number | Depth (cm) | Reported $^{14}$C age (yr BP) | error ± (yr) | 1 sigma age range (cal yr BP) | 2 sigma age range (cal yr BP) | Median probability age (cal yr BP) | Material dated |
|---|---|---|---|---|---|---|---|
| KIA 51561 | 3 | 715 | 20 | 145 – 168[a]<br>172 – 264[b] | 67 – 297 | 201 | Mixed benthic foraminifera |
| KIA 51562 | 103 | 1417 | 29 | 767 – 883 | 709 – 917 | 818 | Mixed benthic foraminifera |
| KIA 51563 | 203 | 1909 | 21 | 1271 – 1350 | 1231 – 1404 | 1313 | Mixed benthic foraminifera |
| KIA 51564 | 303 | 2179 | 28 | 1543 – 1666 | 1484 – 1735 | 1604 | Mixed benthic foraminifera |
| KIA 51565 | 403 | 2554 | 26 | 1982 – 2110 | 1904 – 2174 | 2045 | Mixed benthic foraminifera |
| KIA 51566 | 503 | 2819 | 22 | 2310 – 2424 | 2266 – 2536 | 2373 | Mixed benthic foraminifera |
| KIA 51567 | 603 | 3103 | 22 | 2698 – 2772 | 2659 – 2843 | 2738 | Mixed benthic foraminifera |
| KIA 51568 | 703 | 3509 | 23 | 3168 – 3301 | 3096 – 3351 | 3228 | Mixed benthic foraminifera |
| KIA 51569 | 803 | 3763 | 26 | 3451 – 3571 | 3390 – 3635 | 3516 | Mixed benthic foraminifera |
| KIA 51570 | 903 | 4156 | 25 | 3951 – 4093 | 3876 – 4161 | 4024 | Mixed benthic foraminifera |
| KIA 51571 | 1003 | 4795 | 25 | 4820 – 4925 | 4791 – 5012 | 4877 | Mixed benthic foraminifera |
| KIA 51572 | 1103 | 5572 | 31 | 5743 – 5868 | 5672 – 5908 | 5801 | Mixed benthic foraminifera |

Relative probability: a = 0.192, b = 0.808, according to Calib7.1.

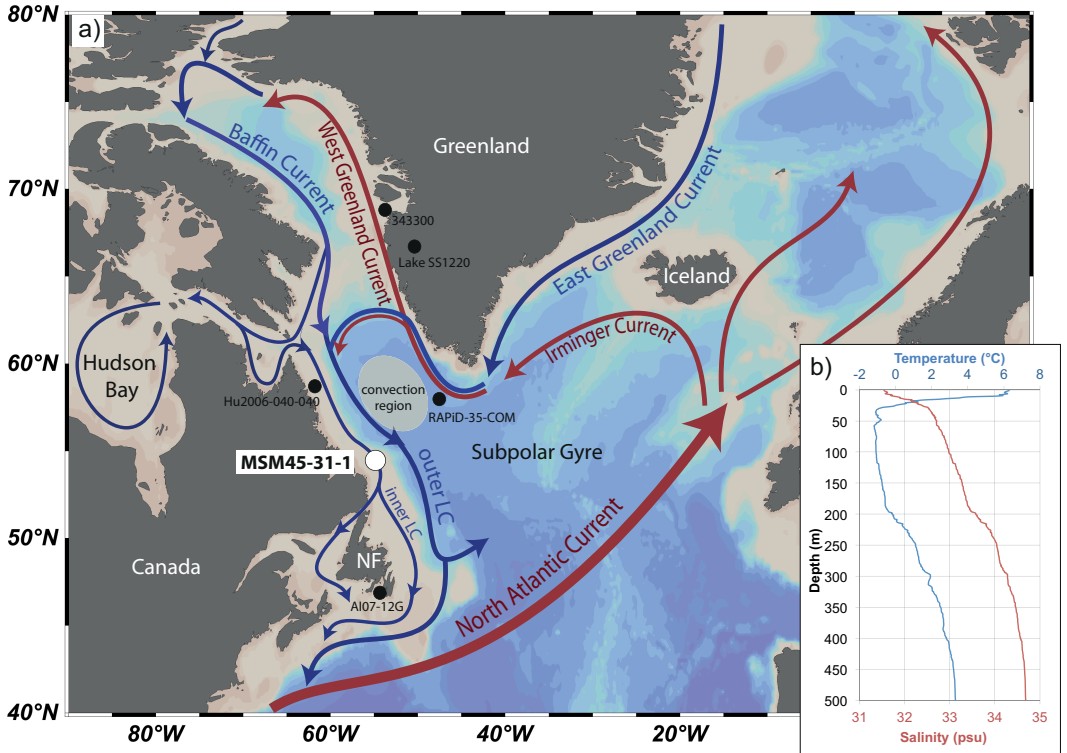

**Figure 1: a)** Map of the North Atlantic showing modern surface circulation with cold (blue) and warm (red) currents (adopted from Lazier and Wright, 1993; Rhein et al., 2011). A shaded area indicates the convection region of LSW. Core MSM45-31-1 is marked by a white dot, other cores referred to are marked by black dots. NF = Newfoundland. **b)** Depth profiles of temperature (blue) and salinity (red) of the water column near the core site at 54°28.53N, 56°04.33W obtained from CTD deployment during cruise MSM45 in August 2015.

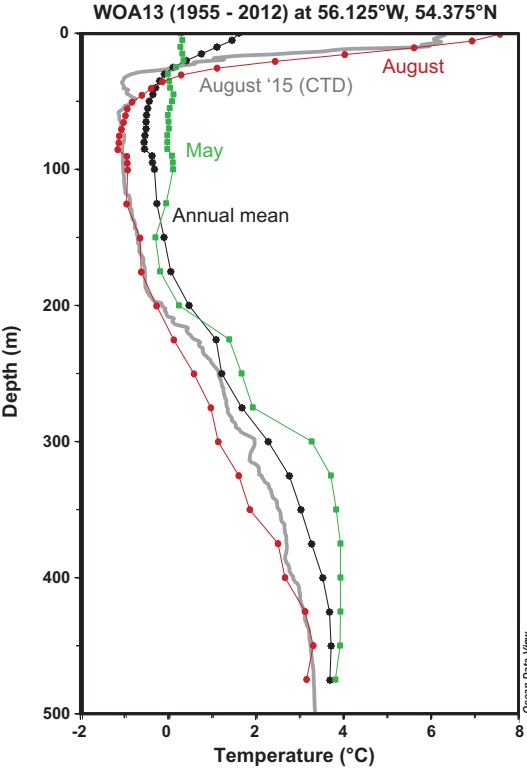

**Figure 2: Comparison of different temperature profiles of the water column near the studied core site. The grey line shows the water profile at 54°28.53N, 56°04.33W obtained during cruise MSM45 in August 2015. Coloured lines mark the profiles at the nearest location at 54°37.50N, 56°12.50W, obtained from the World Ocean Atlas 2013 (WOA13; 1955 – 2012; Locarnini et al.,**

5  **2013), showing annual mean (black), August (red) and May (green) temperatures. August data obtained during MSM45 and from WOA13 show similar temperature profiles with a warming of up to 6 – 8°C at the sea surface. The upper 50m of the water column reflect atmospheric temperatures and therefore differ in the August, May and Annual mean profiles from WOA13. All profiles show temperatures of -1 to 1°C between ca. 40 – 200 m, reflecting the cold, inner LC, and higher temperatures below 200m depth, rising up to 4°C at the shelf bottom, which reflect the influence of the warmer, outer LC strongly influenced by the WGC.**



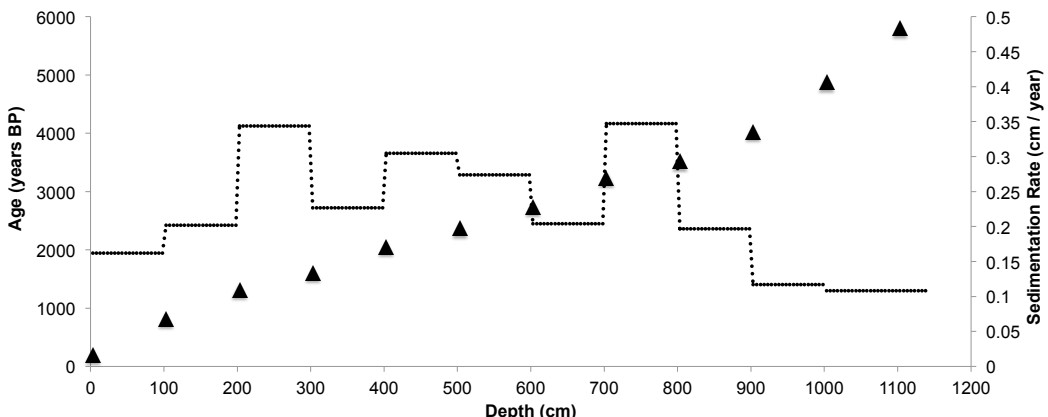

**Figure 3: Age-depth model of core MSM45-31-1. Black triangles mark the calibrated ages of the AMS radiocarbon dates. The black dotted line marks the sedimentation rate (cm per year).**

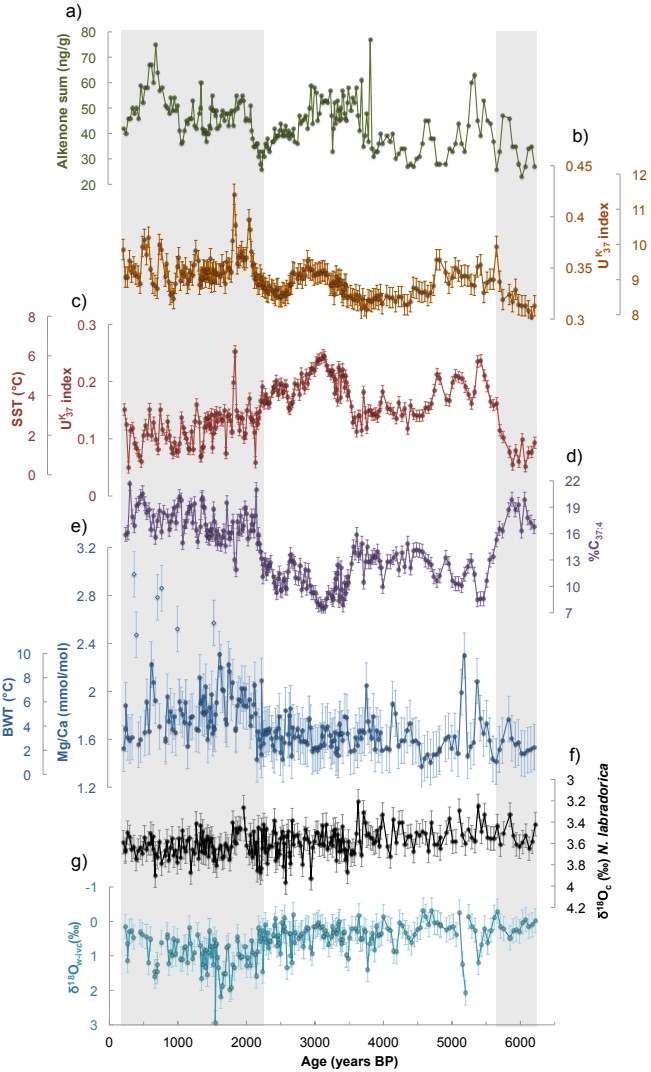

**Figure 4: Down-core record of MSM45-31-1 showing a) alkenone sum, b) $U^{K'}_{37}$ index displayed with 0.01 uncertainty and respective SST estimates, c) $U^{K}_{37}$ index displayed with 0.01 uncertainty and respective SST estimates, d) %$C_{37:4}$ displayed with 0.8 % uncertainty, e) Mg/Ca (mmol/mol) of *N. labradorica* displayed with 0.19 mmol/mol uncertainty and respective BWT estimates, f) $\delta^{18}O_{calcite}$ (‰) of *N. labradorica* displayed with 0.11‰ uncertainty, and g) $\delta^{18}O_{w-ivc}$ (‰) displayed with 0.35‰ uncertainty. Horizontal grey bars mark two cold episodes interrupted by a warm interval.**



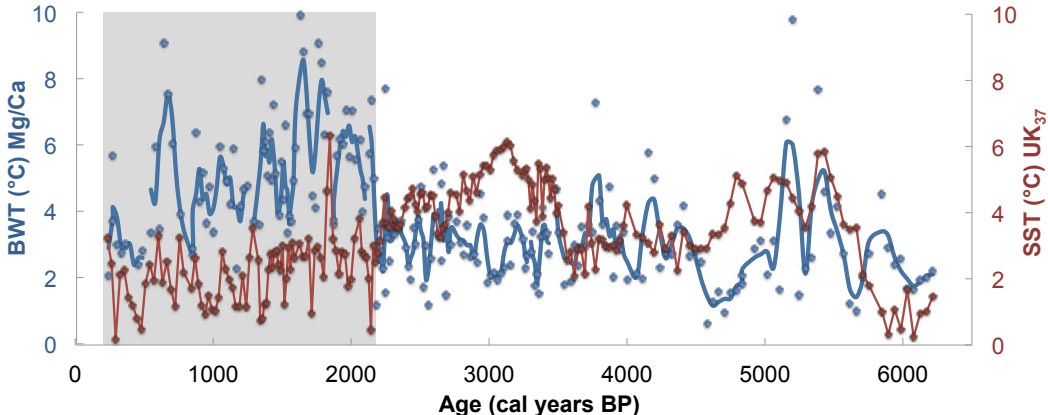

**Figure 5: Surface (red) and bottom (blue) water temperature reconstructions are based on the alkenone U$^{K}_{37}$ index and Mg/Ca ratios of benthic foraminifera *N. labradorica*, respectively. In contrast to the surface record, the BWT record is displayed with a 3-point-runnning mean due to the 2°C uncertainty of individual data points. Both records show a similar temperature range between 6.1 and 2.1 ka BP. After 2.1 ka BP, the BWT record displays higher fluctuations and a shift to warmer average temperatures, leading to a temperature reversal in the water column with generally warmer bottom waters underlying colder surface waters in this interval. The warming of bottom waters is associated with a stronger inflow of the westward retroflection of the WGC reaching our core site at 566m depth underneath the cold inner LC.**

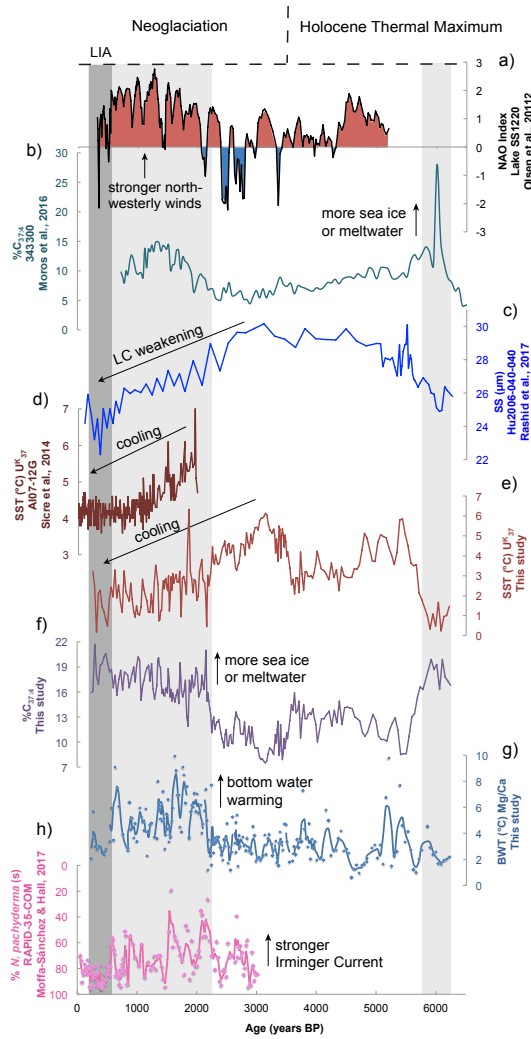

**Figure 6: a) The NAO index (Olsen et al., 2012) with positive (red) and negative (blue) phases. b) %C$_{37:4}$ in core 343300 (Moros et al., 2016). c) SS mean (µm) in core Hu2006-040-040 (Rashid et al., 2017). d) SST record in Newfoundland core AI07-12G (Sicre et al., 2014). e) SST record based on the U$^{K}_{37}$ index (this study). f) Higher proportions of %C$_{37:4}$ are interpreted to reflect a longer sea**

5  **ice season or meltwater. g) BWT estimates are based on Mg/Ca ratios in benthic foraminifera *N. labradorica* and reflect subsurface conditions. h) Lower abundances of polar water planktic foraminifera species *N. pachyderma* in central Labrador Sea core RAPiD-35-COM indicate warming due to stronger IC inflow (Moffa-Sánchez and Hall, 2017).**