# Peer review of "Surface and subsurface Labrador Shelf water mass conditions during the last 6,000 years"

_Climate of the Past, 2019_

## Referee Comment (RC1) · Anonymous Referee #1 · 2 Oct 2019

This study present a paleoceanographic reconstruction of variability in the surface conditions of Labrador Current and changes to the influence from warm Atlantic waters on the subsurface bottom water temperatures over the last 6,000 years. Specifically, a sediment core from the Labrador shelf was analysed; alkenone analysis is applied to reconstruct surface conditions in the Labrador Current, while Mg/Ca measurements are provided to reconstruct the bottom water temperature. The data presented are of high quality given relatively high number of radiocarbon datings as well as the high sedimentation rate and sampling resolution allowing dense subdecadal time series. Moreover, the manuscript is very well written and well structured and the figures, discussion and conclusions easy to follow. I only have a few notions: A particular aim of the study is to assess the impact of Labrador Current variability on Labrador Sea deepwater formation and a fine discussion of mechanisms linking the two is provided. It is found that overall the LC reconstruction here does not match Deep Western Boundary Current reconstruction from south of Newfoundland (Marchitto and de Menocal, 2003). The authors state that more deep water reconstructions are needed to resolve this. In this regard it would have been informative also with a figure of graphs comparing the LC reconstruction presented here with the additional LC reconstructions presented in the manuscript. This would highlight to what extent the surface reconstruction – based on alkenones – are representative of LC variability. Although, as also stated by the authors, similarity between these records (LC SST and DWBC) may reflect that the LC just respond to the same atmospheric forcing that controls deep-water formation, it would still be interesting to understand better exactly how well these data compares to previous LC data from the wider region.

I am not an alkenone expert, but I appreciate the precautions taken in the interpretation of the data, including the caveats associated with C37:4 production from other groups of haptophytes than E. huxlei. This is also evident from the choice of showing and discussing both the UK 37 and UK'37 indexes. I think the authors do a good job of highlighting the issues and I understand the decision of using the UK37 given the similarity with present temperature range in the surface waters. However, I think the manuscript would benefit with adding a discussion of potentially advected alkenones deposited at the core site. I find it interesting that the UK'37 derived temperature, although far higher than SST in the wider region; overall display the same variability as the Mg/Ca bottom temperature reconstruction in the last 6,000 years. Would it be possible that the alkenones deposited at the core site contain a significant fraction of alkenones synthesized far away, in the Irminger Sea (where coccolith blooms are the most extensive for the entire northern North Atlantic Ocean) and are transported with the WGC? The alkenone concentration also resembles the UK'37 and Mg/Ca pattern of variability, which could add support to the reliability of the UK'37 as increased WGC inflow signal and/or increased coccolith blooming in the Irminger Sea? This would not alter the main conclusions of the manuscript, in terms of the paleoceanographic

conclusions arrived at. It would just open up the possibility that alkenones are not only dropped vertically down the water column, but are also transported around the wider ocean with the very dynamic current systems and thus may represent an integrated SST signal from a larger region.

Try not to refer to geographical less known names that are not shown on map (Trinity Bay, Placenta Bay etc. etc)

Inform how far away (km) from the core site the hydrograpic sections (apart from the one measured during the cruise) are located.

---

## Referee Comment (RC2) · Anonymous Referee #2 · 22 Oct 2019

This manuscript uses a multi-proxy approach to reconstruction surface water (alkenone biomarkers) and bottom water (Mg/Ca and stable isotopes) temperatures for the last 6,000 years on the Labrador Shelf and additionally an assessment of the potential role of the Labrador Current (LC) on Labrador Sea convection (LSW formation). Based on these reconstructions, the authors delineate 3 phases of oceanographic conditions in the region. A cold LC phase from 6.2 – 5.6 ka BP, followed by warmer conditions until 2.1 ka BP, which they attribute to the Holocene Thermal Maximum in the region. A water column temperature inversion persisted from 2.1 ka to present, wherein the increasing influence of the warm retroflected West Greenland Current (WGC) and Irminger current (IC) drove warmer bottom water temperature and cooler SSTs and enhanced sea-ice cover were attributed to reduced freshwater supply from a diminished LC. A

shift to a more positive NAO conditions is posited as a cause for these conditions. Mechanisms linking the LC and LSW formation are explored well.

The role of the LC in LSW formation cannot be unequivocally assessed by the data here (nor do the authors maintain that they can), but the manuscript presents a dataset of high quality that focusses particularly on the role of the LC in deep water formation during the mid- to late Holocene, which has received little focus (largely due to a lack of suitable records). Furthermore, the resolution of the marine record allows for sub decadal reconstructions are often lacking in somewhat deeper sedimentary environments. The proxy methods are clearly written and openly discussed: efforts have been made to identify merits/pitfalls of the chosen calibrations and areas of uncertainty (e.g. UK'37 vs UK37 and %C37:4, sections 3.3.2 and 5.1). I would suggest the following minor revisions/additional information before publication:

Sections 3.1 and 3.2 Material and AMS radiocarbon dating. The number of radiocarbon dates is sufficient for the core and an appropriate reservoir correction is used. If the core is, as stated, homogenous then linear interpolation between ages should be sufficient as a more complex age modelling software, e.g. Bacon (Blaauw and Christensen, 2011) would yield a similar age/depth relationship. In this respect, it would be beneficial to add evidence for this homogeneity, such as linescan images, CT scans, grain size etc as support for this, maybe as an item in the supplementary information. If this information has already been published elsewhere it should be referenced.

Section 3.4 Mg/Ca measurements I would include a short explanation as to why (aside from abundance downcore), N. labradorica was chosen for Mg/Ca measurements. E.g. as it is an infaunal species the bottom water carbonate ion concentration has a limited influence and it has, in the Arctic, been deemed a suitable carrier of Mg/Ca in shelf regions (Barrientos et al., 2018). Did the authors consider using the BWT calibration based on the aforementioned paper? As far as I understand it produces estimates in keeping with Skirbekk et al (2016) but extends the temperature range at the cold end of the temperature range? It may not be applicable according to modern BWT (Figures

1 and 2) It would be beneficial to add the depths/intervals at which this 35% of samples may be less accurate (>4 degrees C).

4.1 Chronology State the time interval the core covers (including the +/- for topmost and lowermost age constraints)

Discussion section 5.1 It could be stated again here when discussing BWT that the temperature inversion occurs at the interval in which Mg/Ca estimates are less accurate and should thus be treated with caution (> 4 degrees C).

p 9 line 1: Is Disko Bugt technically Baffin Bay?

p. 9 line 10: Fig. ref should be 6e,f and g?

p9: line 22: in the Moros et al (2016) paper is this not clearer in the dinocyst species (I. minutum %) rather than diatom species at this time?

P. 12 line 28: 'pachyderma'

P. 12 line 29: Figure 6h?

P12 line 31: is this reference to cooling BWTs in the latter part of this interval?

P12 line 31-32: If the previous sentence is referring to cooling BWT temperatures this reference is to subsurface warming? It does not read to clearly and is also mentioned directly in the next paragraph.

P13 line 7: define DWBC as this is the first time it has been mentioned

P13 line 30-31: are there any reconstructions of LSW formation that could be shown on e.g. figure 6?

---

## Referee Comment (RC3) · Anonymous Referee #3 · 25 Oct 2019

This paper presents a high-quality multiproxy record of surface and bottom water conditions at a climate-sensitive location in the subpolar North Atlantic, showing variations in the latter half of the Holocene. The record is very high-resolution, and well-dated, and for that alone is a worthwhile contribution to the field. The discussion and interpretation of the various proxy records are carefully considered and uncertainties and complications are well communicated. I appreciated the discussion of the different alkenone indices and the connection to modern oceanographic data to pick a preferred index, and the caution due to the high concentrations of C37:4 in parts of the record. Overall I think the authors have done an excellent job of situating their record in the context of other records from the subpolar North Atlantic, both in terms of similarities to existing records and in dissimilarities and potential interpretations of those differences.

[Figure]

About the only question I have about the data and interpretation relates to the impact of sedimentation rate changes on the alkenone concentration (e.g. page 4, line 28). It looks to me like the sed rate can vary by up to a factor of 3 based on figure 3 - does this have any effect on the alkenone concentration, can the alkenone concentration be diluted by higher sed rates? I don't think this will change any interpretations - looks to me like the low concentrations at 4-5 ka correspond to low sed rates, so the concentrations aren't low because of dilution, but it might be interesting to calculate some average alkenone fluxes between dated horizons, see how that average flux changes over the course of the record. I don't think it's necessary before the manuscript is accepted, just that I think it would be interesting.

Minor notes:

page 2 last line the current's potential

page 3 line 13 where are these strong north-westerly winds? over the Labrador Sea I assume?

page 3 line 23 itself is formed from the cold...

page 7 line 26 that peaks at 75 ng/g

Figures 5 and 6: mention what the gray bars are in the captions

---

## Author Comment (AC1) · 20 Jan 2020

Thank you very much for the positive and constructive feedback on our contribution. Below, we address the specific concerns raised by reviewer 1.

1. Additional plots of LC reconstructions

As previous LC reconstructions have mainly focussed on late deglacial and early Holocene glacial meltwater run-off (Jennings et al., 2015, Hoffman et al., 2012; Lewis et al., 2012; Hillaire- Marcel et al., 2007; Rashid et al., 2017), only a few records exist that are comparable to our study. Other LC records of the relevant period include a sea-ice dinocyst abundance reconstruction (Solignac et al., 2011) as well as alkenone-based SST reconstructions of the last two millennia from a fjord in Newfoundland (Sicre

et al., 2014). While these datasets indicate similarities to our findings, they do not help to improve our understanding of LC influence on deepwater formation and we therefore refrained from including them in Fig. 6. Following the reviewer's suggestion, a Mg/Ca temperature plot from the Laurentian Slope (Marchitto & DeMenocal, 2003) representing the Deep Western Boundary Current (DWBC) will be added to Fig. 6 (see below). As stated in the original manuscript (p. 13, lines 25 - 31), a Little Ice Age (LIA) cooling is present in, both, DWBC as well as Labrador Shelf bottom waters, suggesting a coupling between the two water masses through the formation of Labrador Sea Water (LSW). However, other significant correlations between our LC reconstruction from the Labrador Shelf and the DWBC record are difficult to identify for the last 4,000 years, probably because of the lower amplitude in bottom water temperature changes in case of the DWBC.

2. Possible advection of alkenones

While it is not inconceivable that alkenones could have been advected from the Irminger Sea, thereby possibly aliasing the temperature signal of surface waters at the core site, there are two lines of evidence that suggest this to be rather unlikely.

a) If the variation in alkenone concentration would indeed reflect changes in alkenone transport from the Irminger Sea by the WGC, we – together with this reviewer - would expect to find a positive correlation between the alkenone concentration and BWT reconstructions, which we infer to reflect the WGC temperature signal. However, these two variables are not correlated (see figure 1 below).

b) Furthermore, the alkenone concentration appears to be independent of the sedimentation rate (see figure 2 below). This excludes the possibility that variations in the sedimentation rate would have impacted the measured fluctuation in alkenone concentration.

3. Try not to refer to geographical less known names that are not shown on map (Trinity Bay, Placenta Bay etc.

In the text, these lesser known names are always used in conjunction with better known and larger scale regional names such as "Newfoundland", or with the core names labelled in Fig. 1, or both. We thus prefer to keep these references to specific bays.

4. Inform how far away (km) from the core site the hydrographic sections (apart from the one measured during the cruise) are located.

This information (27 km) is provided in the revised manuscript "Coloured lines mark the profiles at the nearest location (27 km from the core site) at 54°37.50N, 56°12.50W, obtained from the World Ocean Atlas 2013..."

———————————————————

[Figure]

**Fig. 1.** Our BWT record plotted with an additional temperature record from the Laurentian Slope

R² = 0.04222

**Fig. 2.** Relationship of alkenone sum and BWT

[Figure]

**Fig. 3.** Alkenone sum plotted versus sedimentation rate

---

## Author Comment (AC2) · 20 Jan 2020

Thank you very much for the positive and constructive feedback on our contribution. Below, we address the specific concerns raised by reviewer 2.

1. Add evidence of homogeneity (e.g. linescan images, CT scans, grain size)

We will provide colour scans and core photographs in the appendix.

2. Why was N. labradorica used? Refer to Barrientos et al. 2018. Why was the Barrientos calibration not used?

We will include a brief justification in section 3.4 as to why N.labradorica was used, adopting the wording of this referee: N. labradorica was chosen for Mg/Ca measure-
ments because as an infaunal species the bottom water carbonate ion concentration has been shown to have a limited influence and it has, in the Arctic, been deemed a suitable recorder of Mg/Ca in shelf regions (Barrientos et al., 2018)

We decided not to use the calibration of Barrientos et al. because their data (n=7) are from areas with BWT of less than 0°C, i.e. far below the BWT observed in our study area. In a forthcoming study, we will present new Mg/Ca data of N. labradorica from the Labrador Sea that support the calibration of Skribekk et al. used here.

Add depth intervals of samples >4°C (35%)

See response to (4) below.

3. Chronology: State the time interval the core covers (including the +/- for topmost and lowermost age constraints)

This information will be added.

4. Discussion section 5.1: It could be stated again here when discussing BWT that the temperature inversion occurs at the interval in which Mg/Ca estimates are less accurate and should thus be treated with caution (> 4 degrees C).

Rather than adding this statement to the discussion, we will include the following justification to 'material and methods', following line 8 on page 6: "... As the calibration is based on a temperature range of 1 – 4°C, estimates exceeding 4°C may be less accurate, which is the case for about 35% of the samples reported here. As a result, we refrain from interpreting individual data points exceeding 4°C. However, given the consistency of average Mg/Ca values for different sections of the core, we are confident in our interpretation of changing average BWT for specific time intervals (see 5.2.3 below)

5. p 9 line 1: Is Disko Bugt technically Baffin Bay?

This is correct. We will change the wording to "along the eastern coast of Greenland".

6. p. 9 line 10: Fig. ref should be 6e, f and g?

Correct, this will be changed.

7. p9: line 22: in the Moros et al (2016) paper is this not clearer in the dinocyst species (I. minutum %) rather than diatom species at this time?

The inversed plot of sea-ice dinocyst species I. minutum actually displays a decrease in abundance between 8 and 5.5 ka BP (Moros et al. 2016, Fig. 3), while sea-ice diatom species F. cylindrus indicates a minor increase in abundance at 6.2 ka BP, which we were referring to. However, we agree that this minor increase in F. cylindrus is not sufficient to suggest an increase in sea-ice cover at that time. Therefore, we will change the sentence to: "...they identified a severe drop in warm water benthic foraminifera species I. norcrossi, suggesting an abrupt subsurface cooling (Moros et al., 2016).

8. P. 12 line 28: "pachyderma"

This will be changed.

9. P. 12 line 29: Figure 6h

This will be changed.

10. P12 line 31: is this reference to cooling BWTs in the latter part of this interval?

P12, lines 26 - 32 will be changed to: "Episodes of an increased influx of Atlantic sourced water from the IC were also seen in the central Labrador Sea, indicated by decreases in the abundances of polar water planktic foraminifera species N. pachyderma sinistral (%Nps) at about 2.3 – 2.1 and 1.6 ka BP (RAPiD-35-COM; Moffa-Sánchez and Hall, 2017; Fig. 6h). These quite well correspond to the subsurface warming peaks in our record. Despite the generally warmer conditions in our subsurface water record during the last 2,100 years, the record displays a trend returning to colder temperatures in this interval. A general cooling trend after 2.2 ka BP is also evident in core RAPiD-

35-COM and has been linked to a weakening in LSW formation (Moffa-Sánchez and Hall, 2017).

11. P12 line 31-32: If the previous sentence is referring to cooling BWT temperatures this reference is to subsurface warming? It does not read to clearly and is also mentioned directly in the next paragraph.

We agree that this paragraph is a bit confusing. We now refer to both, a general cooling trend after 2,100 years BP, which is also evident in Moffa-Sanchez and Hall (2017), as well as short episodes of warming observed in our BWT record (Fig. 6g) and in decreases of %N. pachyderma (Fig. 6h).

12. P13 line 7: define DWBC as this is the first time it has been mentioned

"Deep Western Boundary Current" will be added.

13. P13 line 30-31: are there any reconstructions of LSW formation that could be shown on e.g. figure 6?

Unfortunately, no direct reconstructions of LSW formation exist for this period. Therefore, Fig. 6 shows the Irminger Current reconstruction of Moffa-Sanchez and Hall (2017), as circumstantial evidence of LSW formation. Furthermore, following the suggestion of another reviewer, a Mg/Ca temperature reconstruction from the Laurentian Slope (Marchitto & DeMenocal, 2003) representing the Deep Western Boundary Current (DWBC) will be added to Fig. 6 (see figure 1 below).

[Figure]

**Fig. 1.** Our BWT record plotted with an additional temperature record from the Laurentian Slope

---

## Author Comment (AC3) · 20 Jan 2020

Thank you very much for the positive and constructive feedback on our contribution. Below, we address the specific concerns raised by this reviewer.

1. Does the sedimentation rate have any effect on the alkenone concentration?

A comparison of the average alkenone concentration with sedimentation rates between dated horizons shows no correlation (see figures 1 and 2 below). It is thus unlikely that changes in the sedimentation rate caused consistent variations in the alkenone concentrations.

2. page 2 last line the current's potential

[Figure]

This will be changed.

3. page 3 line 13 where are these strong north-westerly winds? over the Labrador Sea I assume?

Correct. This information will be added in the revised manuscript.

4. page 3 line 23 itself is formed from the cold...

This will be changed.

5. page 7 line 26 that peaks at 75 ng/g

This will be changed.

6. Figures 5 and 6: mention what the gray bars are in the captions

This information will be added to the caption: "Grey vertical bars highlight periods of pronounced oceanographic change."

―――――――――――――――――――

[Figure]

**Fig. 1.** Alkenone sum plotted versus sedimentation rate

Fig. 2. Relationship of alkenone sum and sedimentation rate

---

## Author Response (AR1)

**Response to the reviews**

We thank the reviewers for the positive and constructive feedback on our contribution. Below, we address the specific concerns raised by the reviewers point by point and include all relevant changes made to the manuscript.

**1. Additional plots of LC reconstructions**

As previous LC reconstructions have mainly focussed on late deglacial and early Holocene glacial meltwater run-off (Jennings et al., 2015, Hoffman et al., 2012; Lewis et al., 2012; Hillaire- Marcel et al., 2007; Rashid et al., 2017), only a few records exist that are comparable to our study. Other LC records of the relevant period include a sea-ice dinocyst abundance reconstruction (Solignac et al., 2011) as well as alkenone-based SST reconstructions of the last two millennia from a fjord in Newfoundland (Sicre et al., 2014). While these datasets indicate similarities to our findings, they do not help to improve our understanding of LC influence on deepwater formation and we therefore refrained from including them in Fig. 6.

Following one reviewer's suggestion, a Mg/Ca temperature plot from the Laurentian Slope (Marchitto & DeMenocal, 2003) representing the Deep Western Boundary Current (DWBC) was added to Fig. 6 (see below). As stated in the original manuscript (p. 13, lines 25 - 31), a Little Ice Age (LIA) cooling is present in, both, DWBC as well as Labrador Shelf bottom waters, suggesting a coupling between the two water masses through the formation of Labrador Sea Water (LSW). However, other significant correlations between our LC reconstruction from the Labrador Shelf and the DWBC record are difficult to identify for the last 4,000 years, probably because of the lower amplitude in bottom water temperature changes in case of the DWBC.

[Figure]

**2. Possible advection of alkenones**

While it is not inconceivable that alkenones could have been advected from the Irminger Sea, thereby possibly aliasing the temperature signal of surface waters at the core site, there are two lines of evidence that suggest this to be rather unlikely.

A) If the variation in alkenone concentration would indeed reflect changes in alkenone transport from the Irminger Sea by the WGC, we – together with this reviewer -  would

expect to find a positive correlation between the alkenone concentration and BWT reconstructions, which we infer to reflect the WGC temperature signal. However, these two variables are not correlated (see figure below).

[Figure]

B) Furthermore, the alkenone concentration appears to be independent of the sedimentation rate (see figure below). This excludes the possibility that variations in the sedimentation rate would have impacted the measured fluctuation in alkenone concentration.

[Figure]

**3. Try not to refer to geographical less known names that are not shown on map (Trinity Bay, Placenta Bay etc.)**

In the text, these lesser known names are always used in conjunction with better known and larger scale regional names such as "Newfoundland", or with the core names labelled in Fig. 1, or both. We thus prefer to keep these references to specific bays.

**4. Inform how far away (km) from the core site the hydrographic sections (apart from the one measured during the cruise) are located.**

This information (27 km) is provided in the revised manuscript in the caption of Fig. 2: *"Coloured lines mark the profiles at the nearest location (27 km from the core site) at 54°37.50N, 56°12.50W, obtained from the World Ocean Atlas 2013..."*

**5. Add evidence of homogeneity (e.g. linescan images, CT scans, grain size).**

We provide colour scans and core photographs in the supplement.

**6. Why was *N. labradorica* used? Refer to Barrientos et al. 2018. Why was the Barrientos calibration not used?**

We have included a brief justification in section 3.4 as to why *N. labradorica* was used, adopting the wording of this referee (p. 5 lines 26 – 28): "*For Mg/Ca measurements, benthic foraminifera species Nonionellina labradorica was used. Due to its infaunal lifestyle, the bottom water carbonate ion concentration has been shown to have a limited influence and this species has, in the Arctic, been deemed a suitable recorder of Mg/Ca in shelf regions (Barrientos et al., 2018)".*

We decided not to use the calibration of Barrientos et al. because their data (n=7) are from areas with BWT of less than 0°C, i.e. far below the BWT observed in our study area. In a forthcoming study, we will present new Mg/Ca data of *N. labradorica* from the Labrador Sea that support the calibration of Skribekk et al. used here.

**7. Add depth intervals of samples >4°C (35%)**

See response to (9) below.

**8. Chronology: State the time interval the core covers (including the +/- for topmost and lowermost age constraints)**

This information was added (p. 6, lines 29 – 30): *"Based on the lowermost and uppermost age constraints, the core covers at least the period from 5801 (+67/-58) to 201 (+63/-29) years BP."*

**9. Discussion section 5.1: It could be stated again here when discussing BWT that the temperature inversion occurs at the interval in which Mg/Ca estimates are less accurate and should thus be treated with caution (> 4 degrees C).**

Rather than adding this statement to the discussion, we have included the following justification to 'material and methods', following line 9 on page 6: "As the calibration is based on a temperature range of 1 – 4°C, estimates exceeding 4°C may be less accurate, which is the case for about 35% of the samples reported here. *As a result, we refrain from interpreting individual data points exceeding 4°C. However, given the consistency of average Mg/Ca values for different sections of the core, we are confident in our interpretation of changing average BWT for specific time intervals (see 5.2.3 below)."*

**10. P 9, line 1: Is Disko Bugt technically Baffin Bay?**

This is correct. We have changed the wording to "*along the eastern coast of Greenland*" (p 9, line 7 in the new document).

**11. P 9, line 10: Fig. ref should be 6e, f and g?**

Correct, this was changed (p. 9, line 17 in the new document).

**12. P 9, line 22: in the Moros et al (2016) paper is this not clearer in the dinocyst species (*I. minutum* %) rather than diatom species at this time?**

The inversed plot of sea-ice dinocyst species *I. minutum* actually displays a decrease in abundance between 8 and 5.5 ka BP (Moros et al. 2016, Fig. 3), while sea-ice diatom species *F. cylindrus* indicates a minor increase in abundance at 6.2 ka BP, which we were referring to. However, we agree that this minor increase in *F. cylindrus* is not sufficient to suggest an increase in sea-ice cover at that time. Therefore, we have changed the sentence (p 9, lines 28 – 29) to: *"...they identified a severe drop in warm water benthic foraminifera species I. norcrossi, suggesting an abrupt subsurface cooling (Moros et al., 2016)."*

**13. P 12, line 28: 'pachyderma'**

This was changed (p 13, line 3 in the new document).

**14. P 12, line 29: Figure 6h?**

This was changed (p 13, line 4 in the new document).

**15. P 12, line 31: is this reference to cooling BWTs in the latter part of this interval?**

P 13, lines 2 – 8 (new document) was changed to*: "Episodes of an increased influx of Atlantic sourced water from the IC were also seen in the central Labrador Sea, indicated by decreases in the abundances of polar water planktic foraminifera species N. pachyderma sinistral (%Nps) at about 2.3 – 2.1 and 1.6 ka BP (RAPiD-35-COM; Moffa-Sánchez and Hall, 2017; Fig. 6h). These quite well correspond to the subsurface warming peaks in our record. Despite the generally warmer conditions in our subsurface water record during the last 2,100 years, the record displays a trend returning to colder temperatures in this interval. A general cooling trend after 2.2 ka BP is also evident in core RAPiD-35-COM and has been linked to a weakening in LSW formation (Moffa-Sánchez and Hall, 2017).''*

**16. P 12, lines 31-32: If the previous sentence is referring to cooling BWT temperatures this reference is to subsurface warming? It does not read to clearly and is also mentioned directly in the next paragraph.**

We agree that this paragraph is a bit confusing. We now refer to both, a general cooling trend after 2,100 years BP, which is also evident in Moffa-Sanchez and Hall (2017), as well as short episodes of warming observed in our BWT record (Fig. 6g) and in decreases  of %*N. pachyderma* (Fig. 6h).

**17. P 13, line 7: define DWBC as this is the first time it has been mentioned.**

"Deep Western Boundary Current" was added (p 13, line 16 in the new document).

**18. P 13, line 30-31: are there any reconstructions of LSW formation that could be shown on e.g. figure 6?**

Unfortunately, no direct reconstructions of LSW formation exist for this period. Therefore, Fig. 6 shows the Irminger Current reconstruction of Moffa-Sanchez and Hall (2017), as circumstantial evidence of LSW formation. Furthermore, following the suggestion of another reviewer, a Mg/Ca temperature reconstruction from the Laurentian Slope (Marchitto & DeMenocal, 2003) representing the Deep Western Boundary Current (DWBC) was added to Fig. 6 (see answer to (1)).

**19. Does the sedimentation rate have any effect on the alkenone concentration?**

A comparison of the average alkenone concentration with sedimentation rates between dated horizons shows no correlation (see answer to (2)). It is thus unlikely that changes in the sedimentation rate caused consistent variations in the alkenone concentrations.

**20. P 2, last line: the current's potential**

This was changed.

**21. P 3, line 13: where are these strong north-westerly winds? over the Labrador Sea I assume?**

Correct, this information was added in the revised manuscript.

**22. P 3, line 23: itself is formed from the cold...**

This was changed.

**23. P 7, line 26: that peaks at 75 ng/g**

This was changed.

**24. Figures 5 and 6: mention what the gray bars are in the captions**

[revised manuscript text omitted]

---

## Author Response (AR2)

Dear Bjørn Risebrobakken,

Thank you for considering our manuscript for publication in your journal and for your comments regarding the uncertainty of BWT estimates above 4°C. We agree that a reminder of caution is helpful on page 8, line 26-31, and have added a sentence there accordingly. However, we do not think adding a horizontal line at 4°C in the BWT plots would be appropriate. The uncertainty of our estimates increases with temperature, meaning that a value estimated at 4°C may be more accurate than a value estimated at 8°C. This is because the calibration that we used assumes a linear relationship between Mg/Ca ratios and temperature, but an extended temperature range in the calibration data set could eventually show that the relationship is exponential. This would decrease the amplitude of our plot and change the higher absolute BWT values slightly, however the pattern and relative changes in our BWT record would remain the same. In addition, including a line at a specific temperature (4°C) would imply a precise cut-off that is not borne out by the calibration data. As a result, we believe that a horizontal line at 4°C would misrepresent the actual data and the BWT estimate uncertainty. We would thus prefer instead to add another reminder of caution to the figure captions, with reference to the cautionary comment on page 8 (see above).

We hope you understand our concern with your suggestion and remain,

Yours sincerely,
Annalena Lochte et al.